# Causal Discovery in the Presence of Missing Values for Neuropathic Pain Diagnosis

**Ruibo Tu** [* 1]   **Kun Zhang** [* 2]   **Bo Christer Bertilson** [3]   **Clark Glymour** [2]   **Hedvig Kjellström** [1]   **Cheng Zhang** [* 4]

## Abstract

The missing data issue is a common phenomenon in many applications such as healthcare. When applying causal discovery algorithms, such as PC, to a data set with missing values, not properly handling the missing data issue might introduce bias and lead to wrong causal relations. In this work, we identify the potential errors of simply applying PC to data sets with missing values. Further, we extend the constraint-based causal discovery method PC to handle binary data sets with missing values for the neuropathic pain diagnosis[1].

## 1. Introduction

Understanding causal relations from observational data is essential for many machine learning applications, e.g., a healthcare application in (Alaa & van der Schaar, 2017). Many causal discovery algorithms have been developed to determine causal relations from only observational (Glymour et al., 2019). However, these algorithms commonly assume that data are fully observed. Missing data entries are common in many domains and become one of the key practical obstacles that causal discovery methods are facing, especially in the situation when data are not Missing Completely At Random (MCAR). Unfortunately, there are only few works proposing solutions for causal discovery in presence of missing data (Gain & Shpitser, 2018; Strobl et al., 2018). Strobl et al. (2018) use FCI combining with list-wise and test-wise deletion (Spirtes et al., 2000) and output Partial Ancestral Graphs (PAGs) as results which are difficult to interpret, and limit its impact in practical

applications. Gain & Shpitser (2018) propose a PC-based method but assume that the missingness model is known. In this work, we take a real-world neuropathic pain diagnosis application as an example and propose Missing Value PC (MVPC) for binary data. We utilize the MVPC framework that identifies potential erroneous edges by applying PC to the observed part of the data, and propose simple correction methods to recover the true causal graph. Theoretical results on the erroneous edge identification have been presented in our recent work (Tu et al., 2019a). This paper focuses on the new extension of the MVPC framework for binary data with the real-life application to determine the causal relations among pathological labels and symptoms by using neuropathic pain patient record data.

## 2. Method

### 2.1. Preliminaries

**Missingness Graph.** A missingness graph is a causal Directed Acyclic Graph (DAG), $G(\mathbb{V}, \mathbf{E})$ where $\mathbb{V}$ consists of: 1) A set of unobservable nodes $\mathbf{U}$[2]; 2) A set of *substantive* nodes $\mathbf{V}$ containing the set of fully observed variables $\mathbf{V}_o$ and the set of partially observed variables $\mathbf{V}_m$ which is shadowed in gray in our graphical representation; 3) A set of *missingness indicators* $\mathbf{R}$; 4) A set of proxy variables $\mathbf{V}^*$. Figure 1 shows a missingness graph in which $Y$ contains missing values. For the simplicity of our graphical representation, we did not show the proxy variable $Y^*$, an auxiliary variable for the convenience of derivation. $R_y = 1$ means that the value of $Y$ is missing and $Y^*$ takes the missingness value, such as a character "m"; $R_y = 0$ means that the value of $Y$ is observed and $Y^*$ takes the value of $Y$.

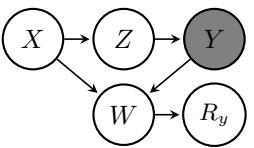

*Figure 1.* A missingness graph

---
[*]Equal contribution   [1]Division of Robotics, Perception, and Learning, KTH Royal Institute of Technology, Stockholm, Sweden [2]Carnegie Mellon University Pittsburgh, PA, USA [3]Karolinska Institute, Stockholm, Sweden [4]Microsoft Research, Cambridge, UK. Correspondence to: Ruibo Tu <ruibo@kth.se>.

*Presented at the first Workshop on the Art of Learning with Missing Values (Artemiss) hosted by the 37[th] International Conference on Machine Learning (ICML). Copyright 2020 by the author(s).*

[1]The long version of the paper is available at https://arxiv.org/abs/1807.04010. The implementation is available at https://github.com/TURuibo/MVPC.

---
[2]In this paper, we assume causal sufficiency. Thus, $\mathbf{U}$ is an empty set.

**Missingness Mechanisms.** All missing data problems belong to one of the three categories (Rubin, 1976): Missing Completely At Random (MCAR), Missing At Random (MAR), and Missing Not At Random (MNAR). Data are MCAR when all missingness indicators are independent of any other variables. Data are MAR when parents of missingness indicators are observed. Data that are neither MAR nor MCAR fall under the MNAR category. Removing samples corrupted by missingness and performing analysis solely with the remaining complete cases will bias conclusions especially when data are MAR or MNAR (Rubin, 2004; Mohan et al., 2013; Shpitser, 2016; Tu et al., 2019a).

**PC and Test-wise deletion PC algorithms.** The PC algorithm (Spirtes et al., 2000) is one of the most commonly used causal discovery methods. Based on Conditional Independence (CI) constraints in data, PC firstly recovers a causal skeleton (undirect causal graph), then orients directions of causal relations, and finally outputs a Completed Partially Directed Acyclic Graph (CPDAG). Test-wise deletion PC algorithm (TD-PC) is the PC algorithm that performs CI tests on those records which do not have missing values for the variables involved in CI tests. Tu et al. (2019a) show that TD-PC can produce erroneous edges when the missingness indicator is the common child or a descendent of the common child.

## 2.2. Missing Value PC for Binary Data

We use an example to introduce how to recover the true causal relations with MVPC for binary data. Suppose that the missingness causal graph of the collected data is shown in Figure 1. According to Proposition 2 in (Tu et al., 2019a), the CPDAG result of applying TD-PC to the data contains an extraneous edge between node $X$ and $Y$. The reason is that when testing whether $X$ and $Y$ are independent given $Z$, we can only get access to the conditional independence relation conditioning on $R_y = 0$ in the data set. In this case, TD-PC implies $X \not\perp\!\!\!\perp Y \mid Z$ regarding the result of the test-wise deletion CI test, $X \not\perp\!\!\!\perp Y^* \mid \{Z, R_y = 0\}$; however, the correct result is $X \perp\!\!\!\perp Y \mid Z$ [3]. Therefore, TD-PC produces an extraneous edge between $X$ and $Y$ because of the wrong implication of the test-wise CI test.

**Binary permutation-based correction.** MVPC with the binary permutation-based correction solves this problem by testing CI relations on generated virtual data whose distribution is $P(X, Y, Z)$. In the generated virtual data, we have the access to the CI relation of $X$ and $Y$ given $Z$. We

---

[3]We denote an independent relation in a data set by " $\perp\!\!\!\perp$ ".

generate the virtual data regarding

$$
\begin{aligned}
P(X, Y, Z) &= \sum_W P(X, Y, Z \mid W) P(W) \\
&= \sum_W P(X, Y^*, Z \mid W, R_y = 0) P(W). \quad (1)
\end{aligned}
$$

Note that we can estimate both of $P(X, Y^*, Z \mid W, R_y = 0)$ and $P(W)$ in Equation 1 with the collected data set. We consider $P(X, Y^*, Z \mid W, R_y = 0)$ as a generator for generating virtual data and $P(W)$ as the input data distribution of the generator. Binary permutation-based correction is summarized in Algorithm 1. Note that conditions of the binary permutation-based correction is that $R_i \perp\!\!\!\perp_d V_i \mid \mathbf{W}$, where $i \in \{x, y, \mathbf{z}\}$ and $X$ denoted by $V_x$ here; and $R_{w_j} \perp\!\!\!\perp_d W_j \mid \mathbf{W}_k$, where $\mathbf{W}_k \subset \mathbf{W} \setminus W_j$.

---

**Algorithm 1** Binary Permutation-based correction

**Input:** data of concerned variables, such as $X$, $Y$, and $Z$ in Figure 1, and the direct causes of their corresponding missingness indicators, such as the direct cause $W$ of $R_y$ in Figure 1.

**Output:** The CI relations among concerned variables, such as the CI relations among $X$, $Y$, and $Z$.

1: Delete records containing any missing value. We denote the deleted data set by $D^d$, and denote the original data set by $D^o$.

2: Split $D^d$ into $D^d_{W=0}$ and $D^d_{W=1}$ according to the values of $W$, e.g., $D^d_{W=0}$ are the samples of $D^d$ in which the value of $W$ is 0.

3: Estimate the joint distributions of $X$, $Y$, and $Z$ with $D^d_{W=0}$ and $D^d_{W=1}$, denoted by $P_{W=0}(X, Y, Z)$ and $P_{W=1}(X, Y, Z)$.

4: Shuffle data of $W$ in $D^o$, denoted by $W^S$, and delete records containing any missing value in $D^o$ (included $W^S$). We denoDte this data set by $D^d_S$.

5: Generate virtual data by sampling from $P_{W=0}(X, Y, Z)$ or $P_{W=1}(X, Y, Z)$ depending on values of $W^S$ in $D^d_S$.

6: Test the CI relations among $\widehat{X}$, $\widehat{Y}$, and $\widehat{Z}$ in the generated virtual data.
**return** The test results of CI relations of $\widehat{X}$, $\widehat{Y}$, and $\widehat{Z}$.

---

**Binary density ratio weighted correction.** When the conditions of binary permutation-based correction cannot be satisfied, we use binary density ratio weighted correction, which gives every data sample a weight according to

$$
\begin{aligned}
P(V) &= \frac{P(\mathbf{R} = \mathbf{0}, V)}{\prod_i P(R_i = 0 \mid Pa(R_i), R_{Pa(R_i)} = 0)} \\
&= P(V \mid \mathbf{R} = \mathbf{0}) \times c \times \prod_i \beta_{Pa(R_i)}, \quad (2)
\end{aligned}
$$

where $V$ is a set of concerning variables, $c = \frac{P(\mathbf{R}=\mathbf{0})}{\prod_i P(R_i=0|R_{Pa(R_i)}=0)}$, and $\beta_{Pa(R_i)} = \frac{P(Pa(R_i)|R_{Pa(R_i)}=0)}{P(Pa(R_i)|R_i=0,R_{Pa(R_i)}=0)}$. Considering the example in Figure 1, the weights can be computed by $\beta_W = \frac{P(W)}{P(W|R_y=0)}$. We firstly delete the records containing any missing value. We denote the samples of the deleted data set in which $W$ takes the value 0 by $D^d_{W=0}$, and the samples in which $W$ takes the value 1 by $D^d_{W=1}$. Next, we test the conditional independence relation of $X$ and $Y$ given $Z$ with the weighted $D^d_{W=0}$ and $D^d_{W=1}$ whose weights are $\beta_{W=0}$ and $\beta_{W=1}$. There are many ways to implement the weighted CI test. For example, the $G^2$ test (McDonald, 2009), a CI test for binary variables, is based on counting the number of times that each possible value combination of $X, Y,$ and $Z$ is observed in a data set. We firstly multiply the counted numbers in $D^d_{W=0}$ by $\beta_{W=0}$ and multiply the counted numbers in $D^d_{W=1}$ by $\beta_{W=1}$. Then, we apply the weighted $G^2$ test.

## 3. Experiment

**Binary synthetic data evaluation.** To evaluate MVPC in the binary case, we used Tetrad (Spirtes et al., 2004) to generate binary datasets together with corresponding ground-truth causal graphs. We compare the performance of baseline methods in MAR and MNAR datasets. The sample size is 600000. The number of substantive variables is 20 in each dataset. The number of missingness indicators is 10 and at most 5 of them producing extraneous edges. We limit the number of the parent of missingness indicators to 1. The values of missingness indicators follow the Bernoulli distribution of which the parameters depend on the parent value of missingness indicators.

We mainly compare MVPC to TD-PC. The *ideal* indicates the result from using fully observed data. As PC has larger sample sizes in CI tests in this case, we also apply PC to the complete data set whose sample size is the average sample size of all CI tests in MVPC/TD-PC. We denote such experiments by *target*. *Target* and *ideal* are targeted ideal performance which is listed as reference. Figure 2 shows that the results of MVPC are clearly better than Test-wise Deletion PC (TD-PC) and close to "target" in the result with Structural Hamming Distance (SHD) metric. Moreover, we find that MVPC with the density ratio weighted correction (MVPC-DRW) in binary cases is much less data-efficiency than MVPC with the permutation-based correction (MVPC-PermC) due to the fact that density ratio weighted correction starts performing better than TD-PC when the samples size is larger than 500000.

**Simulated neuropathic pain diagnosis data evaluation.** We use the recently developed neuropathic pain diagnosis

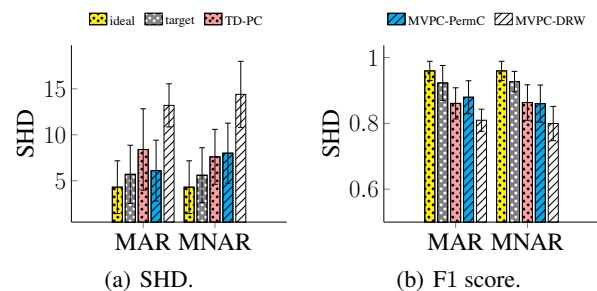

*Figure 2.* Results of the baseline methods in binary cases that are Missing At Random (MAR) and Missing Not At Random (MNAR).

*Table 1.* Results of applying causal discovery methods to simulation data with missing values from the neuropathic pain diagnosis simulator (Tu et al., 2019b).

|  | Cau_acc | Recall | Precision | F1-score |
|---|---|---|---|---|
| PC-ideal | 0.047 | 0.046 | 0.44 | 0.085 |
| PC-target | 0.046 | 0.046 | 0.44 | 0.085 |
| MVPC | 0.045 | 0.043 | 0.452 | 0.078 |
| TD-PC | 0.033 | 0.025 | 0.559 | 0.047 |

simulator (Tu et al., 2019b) to generate patient diagnostic data. The simulator generates diagnostic labels in forms of binary data indicating that certain diagnostic labels exist or not. The ground-truth of causal relations is given thanks to the domain knowledge and the generated data are indistinguishable from real-world patient records verified by the medical expert. We generate 1000 MAR data of 222 variables with the simulator. We evaluate the performance of different methods with the causal accuracy ($Cau\_acc$) (Claassen & Heskes, 2012), recall, precision and F1 score of undirected graphs results. The experiment settings follow (Tu et al., 2019b). The experimental results are shown in Table 1 and Figure 3. We find that the performance of MVPC is close to *target* and *ideal* and better than TD-PC in the simulation experiments.

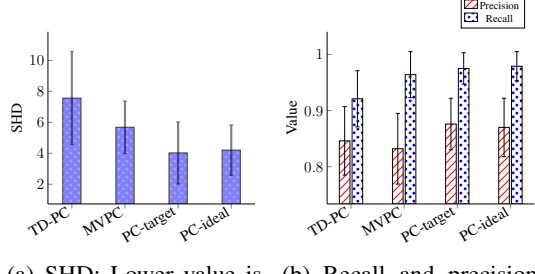

(a) SHD: Lower value is better.

(b) Recall and precision: Higher value is better.

*Figure 3.* Simulation experiments results

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
