# OpenReview forum: "Causal Discovery in the Presence of Missing Values for Neuropathic Pain Diagnosis"
_ICML.cc/2020/Workshop/Artemiss — ICML Artemiss 2020_

### Official Review · AnonReviewer2 · 2020-06-23
**Well structured, but some perspective to usefulness in wider practice would be nice**

**Confidence:** 2
**Rating:** 7

**Review:**

The paper is well written and explains the steps and reasoning behind them nicely. It has a clear structure.

An audience with a background in more classical statistical methods may not be familiar with the abbreviations and algorithms used in the paper. The authors should consider taking that into account when preparing their presentation.

In a previous publication the authors have presented a corresponding approach for continuous data, in the present paper for binary data. In many clinical settings, data will be of mixed type. Can the proposed methodology handle such data?

In the simulated data the authors consider 222 variables. In many clinical applications, a lot fewer variables are measured. How is the proposed method suited to deal with settings with less data, say, 10 to 20 variables?

---

### Decision · Program_Chairs · 2020-07-02

**Decision:**

Accept

**Comment:**

We're happy to accept this paper at Artemiss. We'll contact you soon to inform you about more details concerning the format of your presentation at the workshop, and the camera-ready version deadline. Please take into account the referee's comments to write the camera-ready version.